# A New Approach of Mathematical Analysis of Structure of Graphene as a Potential Material for Composites

**DOI:** 10.3390/ma12233918

**Published:** 2019-11-27

**Authors:** Mieczysław Jaroniek, Leszek Czechowski, Łukasz Kaczmarek, Tomasz Warga, Tomasz Kubiak

**Affiliations:** Faculty of Mechanical Engineering, Lodz University of Technology, 90-924 Łódź, Poland; mieczyslaw.jaroniek@p.lodz.pl (M.J.); leszek.czechowski@p.lodz.pl (L.C.); tomasz.kubiak@p.lodz.pl (T.W.)

**Keywords:** graphene nanostructure, material properties, finite element method, analytical study

## Abstract

The new analysis of a simplified plane model of single-layered graphene is presented in this work as a potential material for reinforcement in ultralight and durable composites. However, owing to the clear literature discrepancies regarding the mechanical properties of graphene, it is extremely difficult to conduct any numerical analysis to design parts of machines and devices made of composites. Therefore, it is necessary to first systemize the analytical and finite element method (FEM) calculations, which will synergize mathematical models, used in the analysis of mechanical properties of graphene sheets, with the very nature of the chemical bond. For this reason, the considered model is a hexagonal mesh simulating the bonds between carbon atoms in graphene. The determination of mechanical properties of graphene was solved using the superposition method and finite element method. The calculation of the graphene tension was performed for two main directions of the graphene arrangement: armchair and zigzag. The computed results were verified and referred to articles and papers in the accessible literature. It was stated that in unloaded flake of graphene, the equilibrium of forces exists; however, owing to changes of inter-atom distance, the inner forces occur, which are responsible for the appearance of strains.

## 1. Introduction

The observed trend in “thinning” the parts of machines and devices as well as structural elements, especially in the aviation or armament industry (unmanned aerial, ground, or underwater vehicles), forces the development of new materials with particular emphasis on composites. Currently, in the world, a lot of attention is devoted to the potential use of graphene as a reinforcement matrix of polymer or polymer–carbon composites. It is assumed that the presence of dispersed graphene flakes will change the mechanics of system cracking. In the case of propagation of cracks and the potential impact of the fracture front with the encountered graphene structure, the energy will be dispersed according to the direction of spatial arrangement of the given graphene flake. This phenomenon can significantly improve the fracture toughness of the entire system, especially when operating in cryogenic conditions, for example, an unmanned aerial vehicle. However, the main problem of the design of composite elements reinforced with graphene is the enormous literature chaos associated with the mechanical parameters of the material, resulting mainly from the fact of projecting the graphene structure onto a lattice model, without taking into account the nature of the chemical bond. This fact undoubtedly makes it impossible to carry out a detailed finite element method (FEM) analysis enabling the design of the morphology and geometry of structural elements. For this reason, we first set up the systematization of mechanical parameters of graphene as an isolated material from the matrix of the composite. 

The two-dimensional nature and carbon hybridization give graphene a unique set of properties, which are electrical [1,2,3], optical [4,5,6], and mechanical [7,8,9,10,11,12,13,14,15,16,17,18,19,20], practically unattainable for other engineering materials. While the optical and electrical research of graphene layers has been mostly perfected, defining the mechanical properties of the material leads to many technical issues, resulting from the nano-scale nature of graphene layers, which makes it impossible to conduct typical tests for the elongation. Any attempt to use nanotest or Atomic Force Microscope (AFM) [21,22,23] to characterize the mechanical properties is also disputable, owing to difficulties with sample mounting, which, in practice, is very difficult and complex. Such a process can also uncontrollably generate structural defects, which makes it very sensitive to mechanical damages [24]. In addition, graphene virtually cannot exist without a carrier in the form of a substrate, which also directly affects the final measurement result. An important factor is also the presence of physical interactions with CO, NO, NO_2_, and water vapor from the atmosphere with a graphene surface, which is not without significance and requires the stabilization of atmospheric composition on a level of ppm [25].

For this reason, a number of scientific centers have attempted to theoretically determine the mechanical properties of graphene [26]. These analyses, owing to the necessity of taking into account the multicriterial model, from both a physical and chemical point of view, are not devoid of defects, which, in the scientific literature, results in a multitude of incomparable results. Meo and Rossi [27] investigated the molecular mechanics of single-walled carbon nanotubes to predict the ultimate strength. Liu et al. [28] analyzed atomic bond breakage using a meshless atomic-scale method. With regard to the literature, many works were devoted to modeling of carbon nanotubes, which can be found in the literature [9,16,21,29,30,31,32,33,34,35,36,37,38,39,40,41,42,43,44,45]. One of them [33] can be here mentioned, where authors dealt with the mechanical properties of single-walled carbon nanotubes (SWCNTs) modeled as Euler beams. On the basis of the elastic deformation energy and mechanical equilibrium of single graphite sheet, they investigated Young’s modulus and Poisson’s ratio of such nanotubes.

The authors of [46,47] also investigated the fracture of nanotubes and a progressive fracture of nanotubes was developed in the work of [48]. From this point of view, the work deals with the simplified plate model of a graphene structure. In many analyses and investigations given in the literature, basic mechanical properties of graphene (Young’s modulus and Poisson’s ratio) were treated as the base for calculations. Both Young’s modulus and Poisson’s ratio in the elastic range are material constants; however, looking through literature, it is unknown how one should assume the values of those parameters for calculations of composite composed of matrixes and flaxes of graphene. As it is well known, graphene, with respect to its mechanical properties, among others, is being researched more and more in applications in the space vehicle and aviation industry, for example, as a structural composite material. However, first of all, it is necessary to systematize the data on the mechanical properties of graphene, which are clearly lacking in the literature. Among the articles concerning graphene, different mechanical properties are given: Young’s modulus Y = 1.2 TPa, 4.70 TPa, 6.6 TPa, 31.7 TPa, or even 86.5 TPa [41]. One should note that Young’s modulus of graphene flaxes treated as orthotropic material (assuming the homogenization) in almost all papers amounts to Y ≅ 1.0 ÷ 1.2 TPa, whereas Young’s modulus of graphene bars Y_Gr_ is characterized by different values. Likely, the values of Poisson’s ratio are different and range, according to the literature, from 0.16 to 0.6. The results in some papers can also be incorrect, for example, in the work of [33], Poisson’s ratio equals 0.2, but following those equations, Poisson’s ratio equals ν_Gr_ ≅ 0.49 ÷ 0.6. For the above reason, the author decide to developed an analytical model based on the theory of elasticity, assuming the incompressibility conditions that are used in continuity mechanics. The proposed model was validated by numerical calculation (finite element method software).

## 2. Assumptions for Modeling the Graphene Structure 

In the present work, the analysis of the plane element of graphene consists of one layer of carbon atoms embedded in polymer matrix. The basic model was hexagonal mesh combining detailed atoms of graphene, which simultaneously are structural nodes (Figure 1). Similar to the classic method of strength analysis, bonds of graphene were stretched (continuous object was compared with the tension of the hexagonal mesh), which allowed the determination of Young’s modulus E_Gr_ and Poisson’s ratio ν for the graphene bar. The model is based on the ideal, chemically pristine structure of graphene, being a single layer of carbon atoms without any structural or chemical defects. We decided that any modelling based on a structure consisting of two or more graphene layers is not justified, as the presence of van der Waals interactions between them dramatically changes the properties of the material.

The analysis of the graphene tension was carried out for two basic directions (if the edge is cut-out along bonds between carbon atoms, one receives the “armchair” edge, otherwise the cutting-out in the perpendicular direction has the edge called “zigzag” [49]).

The model used for calculation was elaborated on the basis of the graphene structure given in other papers [27,29,32,50], among others (Figure 2). Young’s modulus of microstructure was assumed as Y = 1.15 TPa = 11.5 nN/Å^2^ (as in the works of [22,23,32,36,38]). The dimensions of the hexagonal mesh of graphene are presented in Figure 3; that is, l = 1.42 Å, a = 2.46 Å [50]. It was assumed that the thickness of the graphene sheet is the same as the diameters of carbon atom; however, different diameters were also taken into account. For comparison, the computations were conducted for three thicknesses of the graphene sheet (h = 0.44 Å; 0.75 Å; 0.89 Å and d = 0.44 Å; 0.75 Å; 0.89 Å, respectively). In the present work, it was assumed that the hexagonal structure can be divided into repeatable, “Y”-shaped elements interacting with each other through reactions (Figure 2a). For the simplified model, forces and strains were determined through tensions of the considered model. The basic literature states that graphene is a two-dimensional rhomb-shaped carbon mesh with a two-atom basis (Figure 1a). 

Young’s modulus determined on the basis of data from the literature [16,19,21,25,27,32] for the continuous structure of graphene, Y= 1.15 TPa [16], but Young’s modulus for the polymer matrix amounts to E_m_ = 3.5 GPa. Thus, with respect to large difference in stiffnesses, the stresses carried by the matrix, this parameter can be omitted (ratio of Young’s moduli of matrix to graphene is E_m_/Y = 0.003). The tension stresses of elementary distance with dimensions a, h and 3/4l amount to σ_x_ = $\varepsilon_{y}^{\left(1\right)}*Y$ = 1.15 GPa, but the total force F = F_x_ acting in the repeatable element of structure is determined by the following:(1)$$F_{x}=\varepsilon_{x}^{\left(1\right)}\cdot Y\cdot a\cdot h.$$ with the assumptions of average Young’s modulus of the considered part of graphene Y_ave_ = Y, one can calculate the force $F_{x}$ affecting repeatable elements of the structure. For example, for strain $\varepsilon_{Y}^{\left(1\right)}$ = 0.001, the Y = 1.15 TPa, a = 2.46 Å, and h = 0.75 Å force is as follows:(2)$$F_{x}=\varepsilon_{x}^{\left(1\right)}\cdot Y\cdot a\cdot h=2.122\cdot 10^{-2}nN.$$

## 3. Basic Model of Graphene Cell (Calculations of Strains) 

To simplify, substituting continuous elementary segments of graphene by the bar system, it was assumed that the schematic model includes interactions between atoms in the graphene owing to tensions and bending (Figure 4). The shown segment of graphene in the Y-shape can be treated as a macroelement of the graphene structure.

The elongations of system owing to tension are as follows:(3)$$\Delta L=\frac{F_{x}\cdot l}{2E_{Gr}\cdot A};\;\Delta N=\frac{F_{x}\cdot l/2}{4E_{Gr}\cdot A};\;\Delta L+\frac{\Delta N}{2}=\frac{9F\cdot l}{16E_{Gr}\cdot A}.$$

The elongations of system owing to bending can be written as follows:(4)$$f_{b}=\frac{F_{x}\cdot \sqrt{3}}{4}\frac{{\left(\frac{l}{2}\right)}^{3}}{3\cdot E_{Gr}\cdot J}=\frac{F_{x}\sqrt{3}}{4}\cdot \frac{l^{3}}{24E_{Gr}\cdot J}\;;\;f_{s}=\psi \left(1+\nu \right)\frac{F_{x}\sqrt{3}}{4}\cdot \frac{l}{E_{Gr}\cdot A};\;f_{c}=f_{b}+f_{s}=\frac{F_{x}\sqrt{3}}{4}\left(\frac{l^{3}}{24E_{Gr}\cdot J}+\psi \left(1+\nu \right)\frac{l}{E_{Gr}\cdot A}\right),$$where A and J represent the cross-section area and moment of inertia of the graphene bar, respectively. The magnitude ψ denotes the coefficient regarding the influence the shear stresses during strains of the graphene bars (ψ = 1.2). Taking into account both effects, total elongation can be written as follows:(5)$$\Delta l_{c}=\Delta L+\frac{1}{2}\Delta N+\frac{\sqrt{3}}{2}f_{c}=\frac{9F\cdot l}{16E\cdot A}+\frac{3F}{8}\left(\frac{l^{3}}{24E\cdot J}+\psi \left(1+\nu \right)\frac{l}{E\cdot A}\right).$$

The strain in the x-direction of the segment caused by bending and tension is expressed as follows:(6)$$\varepsilon_{x}^{\left(1\right)}=\frac{4}{3l}\left(\Delta L+\frac{1}{2}\Delta N+\frac{\sqrt{3}}{2}f_{c}\right).$$

By transforming the strain in the x-direction, one obtains the following:(7)$$\varepsilon_{x}^{\left(1\right)}=\frac{F_{x}}{4E_{Gr}}\left(\frac{l^{2}}{12J}+\frac{3+2\psi \left(1+\nu \right)}{A}\right).$$

The elongation of the segment of the graphene structure at the length of L_s_ = ¾ l is as follows:(8)$$\Delta l_{c}=\varepsilon_{x}^{\left(1\right)}\cdot L_{s}=\frac{F_{x}}{4E_{Gr}}\left(\frac{l^{2}}{12J}+\frac{3+2\psi \left(1+\nu \right)}{A}\right)\cdot L_{s}.$$

The strain of the alternative segment of the graphene structure with dimensions a, h and length L is as follows:(9)$$\varepsilon_{x}^{\left(2\right)}=\frac{F_{x}}{Y\cdot a\cdot h}.$$

With the assumption of the bar structure of graphene and substitution of the repeatable element of structure by rectangular shape, one can compare both strains $\varepsilon_{Y}^{\left(1\right)}=\varepsilon_{Y}^{\left(2\right)}$. It was assumed that the strains of repeatable of the graphene structure are the same as the continuous structure of graphene with dimensions a, h and length L_s_. Young’s modulus was taken from the literature [22,32,38] as E_Gr_ = Y = 1.15 TPa = 11.5 nN/Å^2^. Hence, substituting Young’s modulus of graphene is determined as follows:(10)$$E_{Gr}=\frac{Y\cdot a\cdot h}{4}\left(\frac{l^{2}}{12\cdot J}+\frac{3+2\psi \left(1+\nu \right)}{A}\right).$$

In this same way as above, the strain in the perpendicular direction was obtained:(11)$$\varepsilon_{y}^{\left(1\right)}=\frac{F_{x}}{4E_{Gr}}\left(\frac{l^{2}}{12\cdot J}+\frac{2\psi \left(1+\nu \right)-1}{A}\right).$$

On the basis of Equations (7) and (11), the substitute Poisson’s ratio of hexagonal structure of graphene can be calculated as follows:(12)$$\nu_{Gr}^{\left(1\right)}=\frac{\varepsilon_{x}^{\left(1\right)}}{\varepsilon_{y}^{\left(1\right)}}.$$

By transforming Equation (11), the force of repeatable element of structure for given strain can be calculated as follows:(13)$$F_{x}=\frac{4\varepsilon_{x}^{\left(1\right)}\cdot E_{Gr}}{\frac{l^{2}}{12J}+\frac{2\psi \left(1+\nu \right)+3}{A}}.$$

## 4. Numerical Model of Graphene Band 

The numerical model of graphene was discussed in many works [28,30,32,47], among others. For the aim of the verification of the assumed macroelement of graphene structure shown in Figure 4, numerical calculation using the finite element method (ANSYS v.18.2 software—Figure 3) was executed.

The mesh of the graphene structure was created by a two-node element beam. The dimensions of numerical model were L_x_ = 14.91 Å, L_y_ = 14.76 Å, and different diameters of bar (see results in Table 1). Numerical models of the segment of the graphene band subjected to tension in the x-direction and y-direction were displayed in Figure 5a (armchair) and Figure 5b (zigzag), respectively.

## 5. The Strain in y-Direction 

Treating the segment of graphene as the basic element for the analysis of forces and strains of the structure, calculations of strains for tension in the y-direction were conducted. In this case, the band of graphene has the “zigzag”-type edge. Owing to the fact that graphene is an anisotropic structure, analogous analysis of strains, as done previously, but in the y-direction, was carried out. The scheme of acting forces on the repeatable element of the graphene structure is displayed in Figure 6a.

The strain for load in the y-direction, perpendicular to the x-axis, is presented in Figure 7b. The lines associating atoms passing through the line are called zigzags (Figure 6b). Initially, the simplified estimations were performed, provided that forces in horizontal elements are forces between atoms. Thus, their influence on strains in the perpendicular direction, with respect to the direction of load, can be defined on the basis of the strain of the graphene band. The elongation of the cut-out structure at the length of $L_{y}=\frac{l\sqrt{3}}{2}$ amounts to $\Delta l_{y}=\varepsilon_{y}\cdot L_{y}$. However, the strain of the substitute segment with dimensions *b*, *h* and *L_y_* is described as follows:(14)$$\varepsilon_{y}^{\left(3\right)}=\frac{F_{y}}{Y\cdot b\cdot h}.$$

With assumption of bar structure of the graphene and substitution of the repeatable element of the structure by cubic element, strains of both models were compared, where b = ¾ l. The strain owing to load in the y-direction is given as follows:(15)$$\varepsilon_{y}^{\left(4\right)}=\frac{F_{y}}{2\sqrt{3}\cdot E_{Gr}}\left(\frac{l^{2}}{12\cdot J}+\frac{3+2\psi \left(1+\nu \right)}{A}\right).$$

Further, the strain in the perpendicular direction regarding loading is formulated as follows: (16)$$\varepsilon_{x}^{\left(4\right)}=\frac{F_{y}}{2\sqrt{3}\cdot E_{Gr}}\left(\frac{l^{2}}{12\cdot J}+\frac{2\psi \left(1+\nu \right)-1}{A}\right).$$

Considering tensile element by forces F_y_ and comparing longitudinal and lateral strains, the substitute Poisson’s ratio is determined as follows:(17)$$\nu_{Gr}^{\left(2\right)}=-\frac{\varepsilon_{x}^{\left(4\right)}}{\varepsilon_{y}^{\left(4\right)}}.$$

Young’s modulus in the case of tension in the y-direction was calculated using the following equation:(18)$$\varepsilon_{x}^{\left(4\right)}=\frac{F_{y}}{2\sqrt{3}\cdot E_{Gr}}\left(\frac{l^{2}}{12J}+\frac{2\psi \left(1+\nu \right)-1}{A}\right),$$
(19)$$E_{Gr}=\frac{3l\cdot h\cdot Y}{2\sqrt{3}}\left(\frac{l^{2}}{24J}+\frac{3+2\psi \left(1+\nu \right)}{2A}\right).$$

If Young’s modulus for the bar of graphene is known, the modulus for the sheet of graphene can be calculated as follows:(20)$$Y=\frac{\sqrt{3}\cdot E_{Gr}}{h}\left(\frac{l^{2}}{16J}+\frac{9+6\psi \left(1+\nu \right)}{2A}l\right).$$

By omitting coefficient ψ, identical results as in the work of [33] can be obtained. Coefficient ψ takes into account the influence of shear stress on the value of beam deflection, which represents a model of the chemical bond between carbon atoms [51].

## 6. Results 

On the basis of the above relations (Equation (10); Equation (12), lines 1–7; and Equation (17), lines 8–14), Young’s modulus of the graphene bar E_Gr_ and substitute Poisson’s ratio for the hexagonal structure of graphene were computed. Those values are set out in Table 1. The first seven lines in the table refer to the x-direction of tension (“armchair”), whereas the next seven lines relate to the y-direction of tension (“zigzag”).

For the purpose of comparison, calculations for seven thicknesses of the graphene sheet (Table 1) were executed. On the basis of the above relations (Equation (10) and (12)), Young’s modulus of the graphene bar E_Gr_ and substitute Poisson’s ratio of the hexagonal structure were estimated. These values are included in Table 1. It should be mentioned that, in another paper [18], Poisson’s ratio of graphene was given as 0.06, 0.19, and 0.22–0.55. In the present paper, for all determined cases (FEM), Poisson’s ratio ranges from 0.373 (“zigzag”) to 0.812 (“armchair”), but these values are still significantly greater than 0.2. Thus, it should be verified which factors have an influence on the strain in the perpendicular direction to tension. It is a question of whether forces between atoms blocking excessive narrowing incompatible with real lateral shortening of graphene should be considered. This phenomenon cannot be caused owing to the presence of a matrix, because the ratio of Young’s moduli of both materials amounts to 0.003. For example, the diameter of the graphene bar and thickness of the sheet equals 0.75Å, and Young’s modulus E_Gr_ of elementary segment consisting of the graphene bar amounts to 13.09 TPa. However, the Poisson’s ratio determined according to the above relation (Equation (12) and (17)) is equal to 0.63 and does not correspond to the real value obtained empirically (ν_exp_ = 0.19–0.22). Moreover, this value is greater than 0.5 (limit value assumed usually for a solid). Therefore, it is interesting if the assumed scheme of the graphene band with omitting forces between atoms responds to the real graphene. It can be assumed that, for the decrease of strain in the perpendicular direction, inner forces in the atom structure are responsible. Poisson’s ratios obtained in an analytical and numerical way for some mechanical properties are comparable (cf: Table 1, line 2). Maps of displacements of the graphene mesh are displayed in Figure 7a (x-direction) and Figure 7b (y-direction).

## 7. Strains in Both Directions 

As stated in the previous subsection, obtaining Poisson’s ratio on the level of 0.2, in both a numerical and an analytical way, is impossible. Hence, apparently, the interaction between graphene atoms should be taken into account, which in practise can be very difficult. Using superposition of forces’ action in both directions (Figure 8), the ratio of forces for Poisson’s ratio can be found.

On the basis of the superposition rule, the value of Poisson’s ratio attributable to both tension forces F_y_ and F_x_ simultaneously equals the following:(21)$$\nu_{xy}=-\frac{\varepsilon_{y}}{\varepsilon_{x}}=-\frac{\varepsilon_{y}^{\left(1\right)}+\varepsilon_{y}^{\left(4\right)}}{\varepsilon_{x}^{\left(1\right)}+\varepsilon_{x}^{\left(4\right)}}.$$

After taking into consideration previously derived relations (Equations (7), (11), (15), (18)), the Poisson’s ratio dependent upon tension forces can be written as follows:(22)$$\nu_{xy}=-\frac{-\frac{F_{x}}{2E_{Gr}^{\left(1\right)}}\left(\frac{l^{2}}{12J}+\frac{2\psi \left(1+\nu \right)-1}{A}\right)+\frac{F_{y}}{\sqrt{3}\cdot E_{Gr}^{\left(2\right)}}\left(\frac{l^{2}}{12J}+\frac{2\psi \left(1+\nu \right)+3}{A}\right)}{\frac{F_{x}}{2E_{Gr}^{\left(1\right)}}\left(\frac{l^{2}}{12J}+\frac{2\psi \left(1+\nu \right)+3}{A}\right)-\frac{F_{y}}{\sqrt{3}\cdot E_{Gr}^{\left(2\right)}}\left(\frac{l^{2}}{12J}+\frac{2\psi \left(1+\nu \right)-1}{A}\right)}.$$

Introducing the ratio of forces as follows:(23)$$k_{xy}=\frac{F_{y}}{F_{x}},$$ and by transforming Equation (22), the ratio k_xy_ can be determined as follows:(24)$$k_{xy}=\frac{\sqrt{3}E_{Gr}^{\left(2\right)}}{2E_{Gr}^{\left(1\right)}}\frac{\left(\frac{l^{2}}{12J}+\frac{2\psi \left(1+\nu \right)-1}{A}\right)-\nu_{xy}\left(\frac{l^{2}}{12J}+\frac{2\psi \left(1+\nu \right)+3}{A}\right)}{\left(\frac{l^{2}}{12J}+\frac{2\psi \left(1+\nu \right)+3}{A}\right)-\nu_{xy}\left(\frac{l^{2}}{12J}+\frac{2\psi \left(1+\nu \right)-1}{A}\right)}.$$

For example, to obtain Poisson’s ratio equal to 0.215 in the range of proportionality, and taking the data of graphene, k_xy_ should equal 0.433 (notice the ratio of forces as well). Hence, it can be concluded that it is necessary to consider the interaction between the graphene atoms, introducing additional bars in the mathematical model of graphene, which is the aim of further work of the authors.

## 8. Summary

In the present work, the new simplified model of the plane graphene structure was investigated. In it, numerous analyses and studies included in the literature of basic mechanical properties (Young’s modulus and Poisson’s ratio) of graphene were presented. The basic model that was used was a hexagonal mesh associated with detailed atoms of graphene, which were simultaneously nodes. The issue was solved on the basis of the superposition method. As in the classic method of the strength analysis, the easy tension of the graphene band (continuous medium was compared with the tension of the hexagonal mesh) was considered to determine the mechanical properties of bars of the mesh. The study of tension was performed for two basic directions (for the “armchair” and “zigzag” direction). It turned out that, for the analytical and numerical calculations, the relations between Young’s moduli of continuous graphene structure and hexagonal structure relate to homogenization principals established in the theory of orthotropic medium. However, the value of Poisson’s ratio of hexagonal mesh does not correspond to values of 0.06, 0.19, and 0.22–0.55, approved in the literature [18]. Also, in the work of [18], it was stated that Poisson’s ratio of graphene can be greater and close to 1.0; therefore, the value of 0.6 is not the special case. Nevertheless, the presented model for calculation should be modified to respond to the real graphene structure. In the unloaded flax of the graphene structure, equilibrium between atoms exists. During tension, owing to changes in the distance between atoms, the forces of atoms’ interaction take different values and have an influence on the strain of graphene. In the case of nanotubes, the graphene mesh constitutes the closed circumference, which essentially disturbs the determination of the forces between atoms. However, in the case of the tensed band on the edges, non-zero forces act. The load by those forces on the edges of the bands of graphene enables the simplified analysis of strain of graphene structure. The values of those forces can be approximately determined for stretched segment along the zigzag edge. The issue concerning mathematical analysis of interaction between graphene atoms requires deeper research and will be a subject of further work of authors, especially with regard to graphene as a reinforcement of composite materials, in the case in which it will also be necessary to consider physicochemical interactions at the boundary of polymer matrix and a reinforcement in the form of graphene.

## Figures and Tables

**Figure 1 materials-12-03918-f001:**
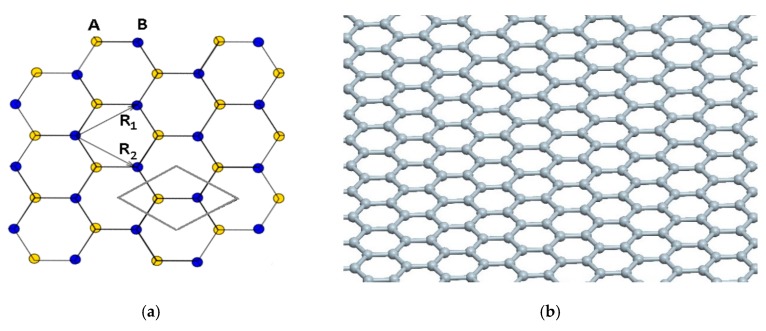
(**a**) Hexagonal mesh of graphene. The elementary cell includes two atoms of carbon; (**b**) Model of graphene given in the works of [5,50], among others.

**Figure 2 materials-12-03918-f002:**
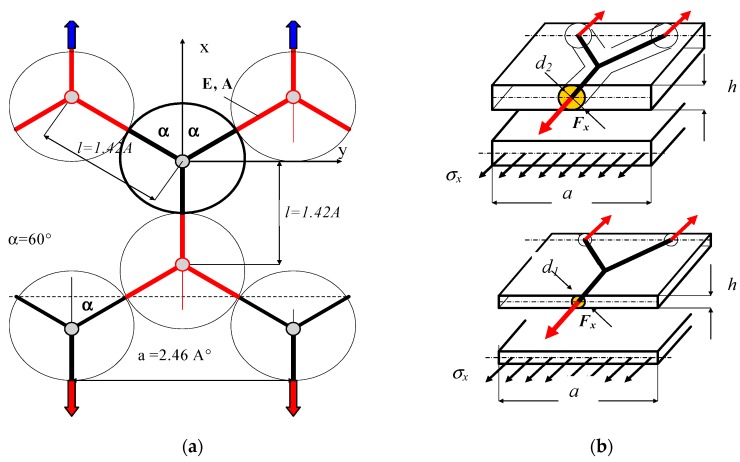
(**a**) Schematic bar system and microelement taken into calculations and (**b**) examples of the graphene element with different thicknesses “h” and diameters “d”. If the edge is cut-out along bonds between carbon atoms, one receives the “armchair”.

**Figure 3 materials-12-03918-f003:**
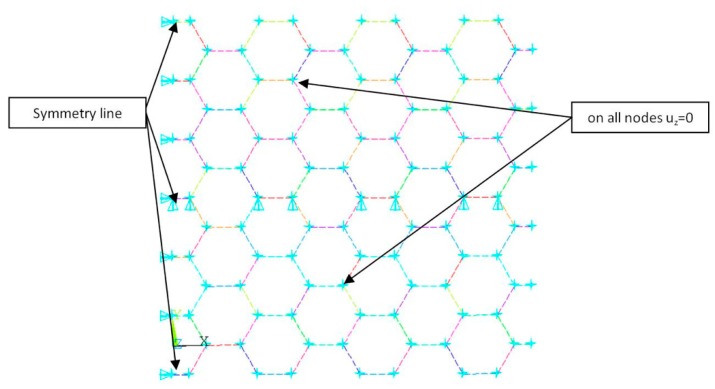
Numerical model of graphene segment prepared for simulation.

**Figure 4 materials-12-03918-f004:**
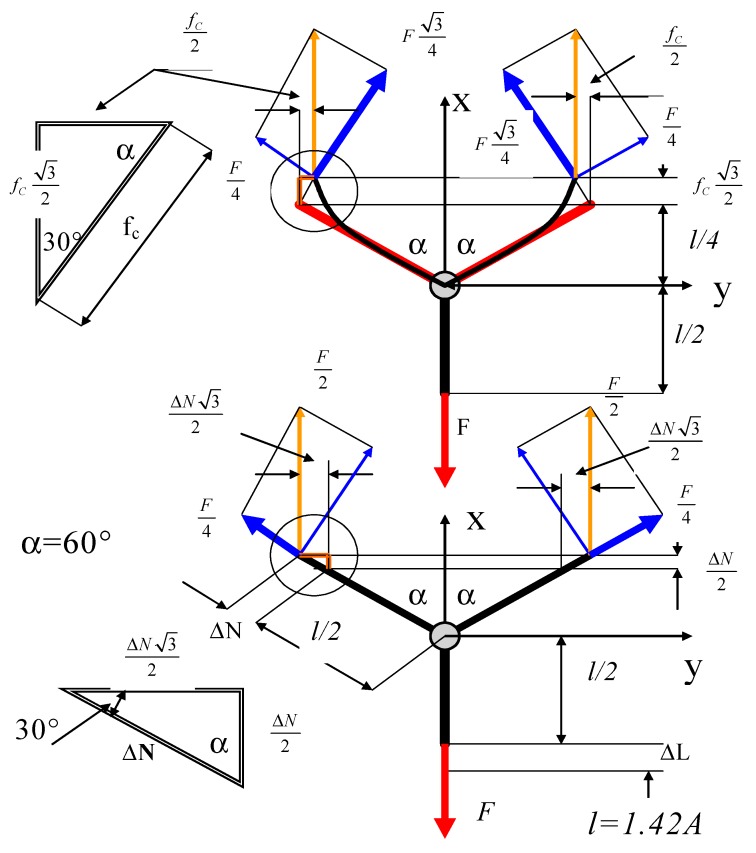
Forces and elongations of bars caused by bending and tension.

**Figure 5 materials-12-03918-f005:**
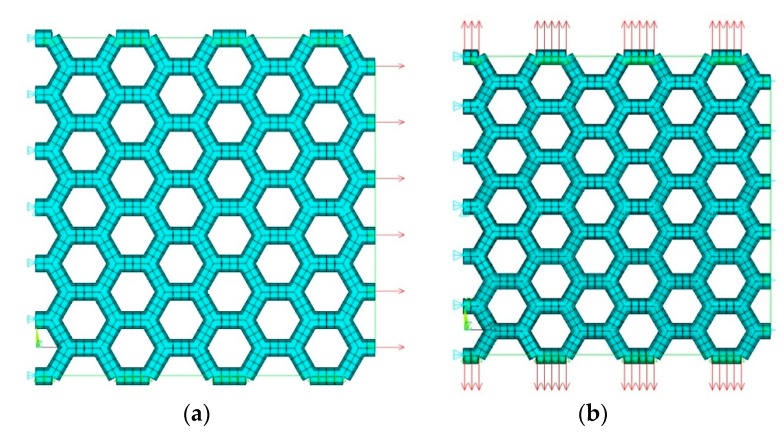
Numerical model of graphene segment prepared for simulation for tension in the x-direction (**a**) (“armchair”) and (**b**) y-direction (“zigzag”).

**Figure 6 materials-12-03918-f006:**
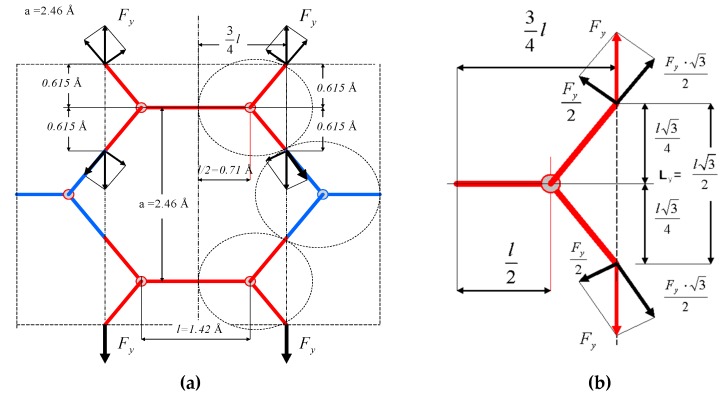
The segment of the (**a**) graphene mesh and (**b**) elementary part for tension in the y-direction. The cut-out in the perpendicular direction has the edge called a “zigzag”.

**Figure 7 materials-12-03918-f007:**
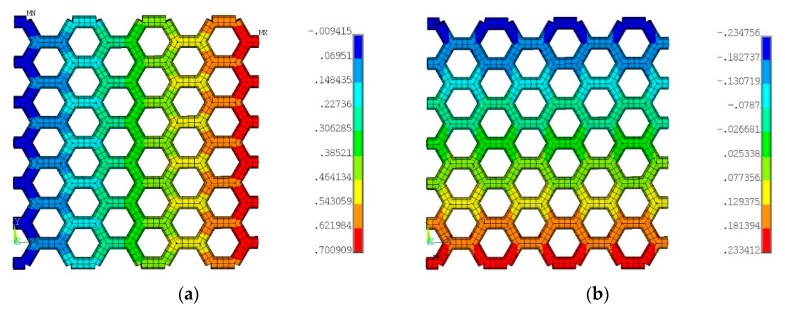
Maps of displacements in the (**a**) x-direction and (**b**) y-direction.

**Figure 8 materials-12-03918-f008:**
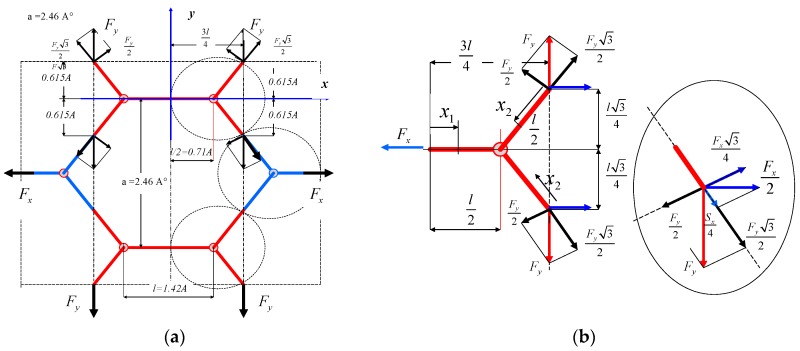
The segment of the graphene mesh with (**a**) two perpendicular forces and (**b**) the elementary part for tension in the x- and y-directions.

**Table 1 materials-12-03918-t001:** Results of calculations. FEM, finite element method.

Number of Calculation Variant	Direction of Tension	l (Å)	a (Å)	h (Å)	d (Å)	Y (Tpa)	ν (-)	E_Gr_ (TPa) (Equation (10))	$\nu_{Gr}$ (-) (Equation (12)) (Equation (17))	$\nu_{Gr}$ (-) FEM
1	x	1.42	2.46	0.44	0.44	1.15	0.3	40.95	0.800	0.812
2	x	1.42	2.46	0.75	0.75	1.15	0.3	13.09	0.663	0.666
3	x	1.42	2.46	0.89	0.89	1.15	0.3	9.630	0.580	0.620
4	x	1.42	2.46	1.00	1.00	1.15	0.3	7.932	0.546	0.59
5	x	1.42	2.46	1.42	1.42	1.15	0.3	4.727	0.463	0.515
6	x	1.42	2.46	2.00	2.00	1.15	0.3	3.058	0.411	0.471
7	x	1.42	2.46	2.42	2.42	1.15	0.3	2.448	0.392	0.453
8	y	1.42	2.46	0.44	0.44	1.15	0.3	40.94	0.800	0.791
9	y	1.42	2.46	0.75	0.75	1.15	0.3	13.08	0.633	0.618
10	y	1.42	2.46	0.89	0.89	1.15	0.3	9.620	0.580	0.564
11	y	1.42	2.46	1.00	1.00	1.15	0.3	7.930	0.546	0.527
12	y	1.42	2.46	1.42	1.42	1.15	0.3	4.726	0.463	0.445
13	y	1.42	2.46	2.00	2.00	1.15	0.3	3.057	0.411	0.394
14	y	1.42	2.46	2.42	2.42	1.15	0.3	2.448	0.392	0.373

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
