# Peer review of "A New Approach of Mathematical Analysis of Structure of Graphene as a Potential Material for Composites"

_materials, 2019, doi:10.3390/ma12233918_

Round 1
Reviewer 1 Report
This manuscript gave a method for determining the mechanical properties of graphene materials, especially Poisson’s Ratio. This method is simple and straightforward. Nevertheless, it is not easy to determine how close these calculations are to real numbers. The following comments are suggested to improve the manuscript. Lack of explanation of ‘FEM’. Please explain the definition and difference of ‘armchair’ and ‘zigzag’ more clearly, i.e. labels in Figure 7. Line 164-166, equations are unclear. Line 178, what is the physical meaning of omitting ψ? Please provide more details about reference 34. Why or why can’t modeling methods for carbon nanotubes be used for graphene? Defects are often unavoidable in graphene materials, how to correct these effects in this model?Author Response
Please see the attachment.

Reviewer 2 Report
For authors - it must be improved the presentation in Figures and make some corrections in the text (more clear explanation in some points - see in attched file)

Reviewer 3 Report
The authors theoretically study the mechanical response of the finite graphene ribbon under the strain, using the finite element method.
According to the manuscript, they point out the current theoretical controversies in the mechanical property of the graphene flaxes. Authors argue that the mechanical property of graphene in the current literatures has a lack of consensus. However, they do not explain under what conditions these calculations have been performed and why they diverge. Authors need to offer this information to make the manuscript more persuasive rather than mentioning vague future applications of graphene. The overall presentation of the current manuscript is confusing.
In addition, despite these motivations, the analysis of the current manuscript is limited to the single-layer graphene with only two directions of the strain. Under this simplified models, it is very difficult to understand the accomplishment of this paper. Authors need to resolve these issues for publication in any form.
Round 2
Reviewer 3 Report
The authors have answered my questions and criticisms. I recommend the publication for Materials.